# Testing implementation facilitation for uptake of an evidence-based psychosocial intervention in VA homeless programs: A hybrid type III trial

**David A. Smelson**[1,2]*, **Vera Yakovchenko**[1], **Thomas Byrne**[1,3], **Megan B. McCullough**[1,4], **Jeffrey L. Smith**[5], **Kathryn E. Bruzios**[1,2], **Sonya Gabrielian**[6,7]

1 Veterans Affairs Bridging the Care Continuum-Quality Enhancement Research Initiative, Bedford, Massachusetts, United States of America, 2 Department of Psychiatry, University of Massachusetts Medical School, Worcester, Massachusetts, United States of America, 3 School of Social Work, Boston University, Boston, Massachusetts, United States of America, 4 Department of Public Health, Zuckerberg College of Health Sciences, University of Massachusetts Lowell, Lowell, Massachusetts, United States of America, 5 Central Arkansas Veterans Healthcare System, Little Rock, Arkansas, United States of America, 6 Veterans Affairs Greater Los Angeles Health Care System, Los Angeles, California, United States of America, 7 David Geffen School of Medicine, University of California at Los Angeles, Los Angeles, California, United States of America

* David.smelson@va.gov

**Data Availability Statement:** These analyses were performed using VHA data. Deidentified data can

## Abstract

### Background

Healthcare systems face difficulty implementing evidence-based practices, particularly multicomponent interventions. Additional challenges occur in settings serving vulnerable populations such as homeless Veterans, given the population's acuity, multiple service needs, and organizational barriers. Implementation Facilitation (IF) is a strategy to support the uptake of evidence-based practices. This study's aim was to simultaneously examine IF on the uptake of Maintaining Independence and Sobriety Through Systems Integration, Outreach and Networking-Veterans Edition (MISSION-Vet), an evidence-based multicomponent treatment engagement intervention for homeless Veterans with co-occurring mental health and substance abuse, and clinical outcomes among Veterans receiving MISSION-Vet.

### Methods

This multi-site hybrid III modified stepped-wedge trial involved seven programs at two Veterans Affairs Medical Centers comparing Implementation as Usual (IU; training and educational materials) to IF (IU + internal and external facilitation).

### Results

A total of 110 facilitation events averaging 27 minutes were conducted, of which 85% were virtual. Staff (case managers and peer specialists; n = 108) were trained in MISSION-Vet and completed organizational readiness assessments (n = 77). Although both sites reported

be provided upon request pending ethical approval and in accordance with VHA guidelines and permissions. Please contact Dr. John Wells at john.wells5@va.gov or (781) 687-2924.

**Funding:** All the authors are funded by a grant from the Health Services Research and Development Quality Enhancement Research Initiative, "Bridging the Care Continuum" (QUE 15-284).

**Competing interests:** The authors have declared that no competing interests exist.

being willing to innovate and a desire to improve outcomes, implementation climate significantly differed. Following IU, no staff at either site conducted MISSION-Vet. Following IF, there was a significant MISSION-Vet implementation difference between sites (53% vs. 14%, $p = .002$). Among the 93 Veterans that received any MISSION-Vet services, they received an average of six sessions. Significant positive associations were found between number of MISSION-Vet sessions and outpatient treatment engagement measured by the number of outpatient visits attended.

## Conclusions

While staff were interested in improving patient outcomes, MISSION-Vet was not implemented with IU. IF supported MISSION-Vet uptake and increased outpatient service utilization, but MISSION-Vet still proved difficult to implement particularly in the larger healthcare system. Future studies might tailor implementation strategies to organizational readiness.

## Trial registration

ClinicalTrials.gov, NCT02942979.

## Introduction

Large healthcare systems, including the Department of Veterans Affairs (VA), are working towards eliminating care inefficiencies by integrating administrative, operations, and novel clinical interventions [1]. This is relevant to the VA's strategic commitment to end Veteran homelessness, resulting in dramatic expansion in their programs [2]. This expansion included implementing evidence-based practices VA-wide to improve outcomes, which can often prove difficult given the need to broadly address individual (e.g., training, involvement in decision making) and organizational factors (organization size, climate, support for the practices among staff and administrators [3–5]. Evidence-based practices are often challenging to uniformly adopt across a system even when they are known to improve outcomes due to a variety of barriers including time-consuming training requirements or changing provider routines [3]. These barriers can be particularly challenging for programs serving vulnerable populations, such as Veterans experiencing homelessness, given that these programs often address multiple behavioral health, substance abuse, medical care and social client needs simultaneously, while also being mindful of such issues as care fragmentation and treatment engagement [6].

Maintaining Independence and Sobriety through Systems Integration, Outreach and Networking-Veterans Edition (MISSION-Vet) is an evidence-based multicomponent wraparound treatment engagement approach for homeless Veterans with a co-occurring mental health and substance use disorder [7]. While a detailed description of MISSION-Vet is included in the methods section below, in short, MISSION-Vet is a team-based hybrid psychosocial and linkage intervention with the primary objective of engaging homeless Veterans with co-occurring disorders in outpatient care. This intervention aims to address the challenges posed by low rates of treatment engagement for homeless Veterans with co-occurring disorders, as treatment is critical for housing sustainability and recovery [8, 9]. MISSION-Vet has improved treatment attendance and engagement, mental health and substance abuse outcomes, and reduced days homeless [5, 10–14]. MISSION-Vet was also certified by Substance Abuse and

Mental Health Services Administration's National Registry of Evidence Based Practices [5, 10–12]. While MISSION-Vet implementation within VA homeless programs fills a current care gap, as a team-based multicomponent intervention, it also presents implementation challenges [15, 16]. These challenges include MISSION-Vet requiring 1) case managers and peers to work together as a team despite having somewhat different training and philosophies, 2) staff to use as hybrid model, which includes both running psychoeducational groups and doing assertive community outreach, and 3) staff using a stepdown model with decreasing intensity so clients ultimately engage in community supports [17]. Thus, the model intensity and complexity make it more complicated to implement as compared to single discipline and single modality interventions.

A previous study with Getting to Outcomes (GTO), an implementation model focused on capacity building for uptake of MISSION-Vet in VA homeless programs nationally found that while MISSION-Vet was implemented at all the sites, the intervention intensity and complexity, organizational demands, and time required for GTO resulted in lower MISSION-Vet uptake [18]. There are other implementation approaches to help organizations adopt and sustain complex evidence-based approaches like MISSION-Vet [19]. Implementation facilitation (IF) is an evidence-based strategy conceptualized within the integrated 'Promoting Action on Research Implementation in Health Services' (i-PARIHS) framework, in which facilitators external and/or internal to a healthcare organization work to support successful implementation of clinical innovations by assisting stakeholders with planning, execution, and refinement that addresses factors related to: (a) characteristics of the innovation itself; (b) the outer and inner context of the healthcare setting; and (c) characteristics of the recipients of the innovation [20]. IF is multifaceted, involving interactive problem-solving and support that occurs in a context of a recognized need for improvement and supportive interpersonal relationships [19]. Given previous work showing IF to be an effective strategy for implementing complex clinical innovations, we posited that it may be effective for implementing a multicomponent intervention like MISSION-Vet, in a complex healthcare system like VA, and in programs serving a high acuity population of homeless Veterans with a co-occurring disorder [21]. Therefore, consistent with recommendations for Type III hybrid implementation-effectiveness designs [22], the primary aim was to study the impact of the IF strategy to support MISSION-Vet implementation and fidelity. The secondary implementation aim was to examine organizational readiness differences between sites. The effectiveness aim was to examine the association between receipt of MISSION-Vet and treatment engagement in clinical services, as measured by VA inpatient and outpatient service utilization.

## Method

### Study design

This was a multi-site randomized hybrid type III implementation-effectiveness modified stepped-wedge trial in seven homeless programs at two VA Medical Centers (VAMCs) serving homeless Veterans with co-occurring mental health and substance use problems [22]. Hybrid type III trials are intended for interventions like MISSION-Vet with robust effectiveness data; though effectiveness data may be further gathered, the core focus of these trials is implementation of the client level intervention (here, MISSION-Vet). We selected a modified stepped-wedge design to compare MISSION-Vet uptake under two staff level intervention conditions: Implementation as Usual (IU) versus Implementation Facilitation (IF) [22]. This is a modified stepped-wedge design in that, in contrast to a traditional stepped-wedge wherein all sites initially start in the control condition, with programs switching over to the intervention condition at fixed intervals or steps and then remaining in the intervention condition for the

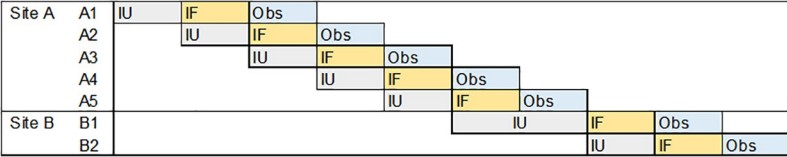

**Fig 1. Modified stepped-wedge trial design.**

duration of the study, in our study, it was not feasible to implement IF at all programs simultaneously. As a result, in the present study, each program received six months of IU and crossed over to IF for another six months, with data collection continuing for an additional six months after the end of IF. Moreover, in terms of calendar time, the rollout of the control and intervention conditions in our study occurred in a sequential manner across programs, with each program being in the intervention and control conditions for a fixed period of time, and not contributing to the analysis in periods in which they are neither in the intervention or control condition (Fig 1). This type of design is akin to what the stepped-wedge design literature refers to as an incomplete stepped-wedge design with limited measurements prior to and after crossover [23, 24] and is appropriate to use when it is not feasible to implement an intervention or collect data in all clusters simultaneously, either due to resource constraints or for other reasons. In this case, we did not have the resources to do IF in every location at the same time. In addition, to further the existing literature supporting MISSION-Vet outcomes, this trial also enabled us to extract existing data from the VA Electronic Medical Record to examine treatment engagement among those receiving MISSION-Vet.

The project was deemed quality improvement and received an exempt status by the Institutional Review Board at the Bedford, Massachusetts VAMC according to the VA Program Guide 1200.21 [25], thus waiving the need for written or verbal informed consent. The staff being trained were the study participants and they were informed about the project's designation as Quality Improvement. Data from the medical record were extracted for the clients being served by the staff members offering MISSION-Vet. Staff were informed of this data extraction and records were not anonymized as it was necessary to identify the clients being served by the staff offering MISSION-Vet.

## MISSION-Vet intervention

A detailed description of MISSION-Vet client level intervention is reported in a previous protocol paper [26]. In brief, MISSION-Vet is delivered by a master's level social work case manager, and a peer specialist, the latter of which is someone with prior lived experience with homelessness, substance abuse and mental health issues. The MISSION-Vet team delivers the following five treatment components: critical time intervention, dual recovery therapy (DRT), peer support, vocational and educational support, and trauma-informed care, all guided by Housing First and harm reduction philosophies that emphasize low barrier services for clients [27–32]. Both the case manager and peer specialist offer psychoeducational sessions either individually or in groups (13 DRT co-occurring disorders groups and 11 peer support recovery groups both designed to empower Veterans to engage in treatment) along with unstructured community outreach sessions to engage clients in care and link Veterans to other needed community support. MISSION-Vet was offered for approximately 2-hours a week for 6-months, and service delivery was guided by a Treatment Manual [7]. Veterans could also receive a MISSION-Vet Workbook that includes assignments reinforcing recovery [33].

## Participants and recruitment

The project was conducted at two VAMCs (hereafter Sites A and B) and offered to staff at seven homeless programs (four locations at Site A and three locations at Site B) with the unit of measurement being the two VAMC's. The two VAMCs were selected because of the size and scope of the healthcare systems, geographic dispersion, and the rate of homelessness in the regions [34]. Site A was in a large VA urban setting, serving approximately 83,000 unique Veterans annually with the highest-level complexity, and two on-site residential buildings, and two off-site buildings. Site B was a smaller suburban medium complexity VA serving approximately 18,000 unique Veterans annually, with one on-site and off-site residential building. Both Sites A and B had community-based non-residential treatment which included housing placement, case management, linkages to mental health and substance use programing, but not MISSION-Vet.

There were two groups of participants in the study: staff and clients. The first group are the case managers and peer specialists delivering MISSION-Vet; staff participation was voluntary with no incentives provided. The second group were, Veterans (clients) being served by these staff in their respective VA homeless programs. Staff were encouraged to follow the recommended MISSION-Vet inclusion and exclusion criteria. This included: (1) enrolled in a VA homeless program at one of the implementation sites; (2) met Diagnostic and Statistical Manual of Mental Disorders, 5th Edition [35] diagnostic criteria or International Classification of Diseases, 10th Revision [36] for current substance use disorder (e.g., alcohol, marijuana, cocaine) and a co-occurring mental illness which includes anxiety, mood, or a psychotic spectrum disorder.

## Implementation strategy

**Implementation as usual (IU).** At the outset, leadership from both VAMCs were introduced to MISSION-Vet and invited to participate. IU was comprised of a 1.5-hour webinar training offered at least twice to case manager and peer specialist staff within the two VAMCs and seven programs to accommodate scheduling. Training provided an overview of the MISSION-Vet approach and the implementation materials (MISSION-Vet Treatment Manual, MISSION-Vet Consumer Workbook, MISSION-Vet Fidelity Measure), and staff roles [7, 33]. It also presented how to use the MISSION-Vet service delivery fidelity measure, embedded within the VA medical record. We used this fidelity measure to capture the total number and type of MISSION-Vet sessions delivered, which served as our measures of MISSION-Vet uptake. This passive implementation strategy has been used in previous studies [37].

**Implementation facilitation (IF).** Following initial training, there was a 6-month waiting period prior to the 6-months of IF being offered to each of the seven programs at the two VAMCs. As noted in Fig 1, IF was turned on in a stepwise fashion at each of the seven sites over a 21-month period, with the sites randomly assigned to a particular step. External facilitation is the form of IF used in this study and delivered by outside IF experts with specialized knowledge of implementation and quality improvement approaches [38–40]. In MISSION-Vet, IF experts partner with facility staff to implement MISSION-Vet through implementation planning, goal-setting and problem-solving [41–43]. Another IF goal was to work with VAMCs to tailor the evidence based practice where appropriate to meet local contextual demands. The external facilitators (JLS, VY) held bi-weekly meetings with local program staff executing MISSION-Vet to address implementation barriers, troubleshoot, and provide implementation fidelity reports, which included feedback on number and type of MISSION-Vet services delivered. External facilitators also provided regular feedback on the staff's use of the

fidelity measure within the VA medical record since this fidelity measure was used to construct our measure of implementation uptake.

## Measures

Project measures captured information about organizational readiness, implementation outcomes (both IF and the implementation of MISSION-Vet), and VA health services utilization. Depending on the outcome, data were measured at the site, staff, and/or veteran level. With regard organizational readiness, we used an abbreviated version of the Organizational Readiness to Change Assessment (ORCA) context subscale and Jacobs' Implementation Climate survey, resulting in a 21-item 5-point Likert scale to get at site and staff level readiness [44, 45]. Higher scores indicate greater organizational readiness and implementation climate. Following MISSION-Vet training, staff were asked to complete the organizational readiness survey and demographic survey regarding their age, sex, role/position, and tenure in VA.

Consistent with recommendations for type III hybrid effectiveness-implementation designs, our primary outcome was MISSION-Vet uptake (as measured by number of MISSION-Vet sessions delivered) during the IU versus IF time periods [22] and the secondary aim was to assess clinical outcomes (health service utilization). For this comparison of IU versus IF timeframes, we used a standardized facilitation tracking sheet completed by the external facilitators at the site level that included date, length of time, parties involved, activity type [46]. In addition, MISSION-Vet implementation was collected with a fidelity measure that was embedded in the Veterans' electronic medical record using a specially created service tracking note template to quantify the type and amount of MISSION-Vet delivered [18]. Information captured in this note template included: which DRT sessions, peer support sessions, and Consumer Workbook exercises were completed; whether the MISSION-Vet Consumer workbook was provided; whether community activities were done with a Veteran (e.g., taken to appointment, NA/AA meetings, meetings with landlords); and referrals made to other services. Finally, treatment engagement as an outcome was captured with Veterans' medical records obtained from the VA Corporate Data Warehouse, which included number of MISSION-Vet contacts and other outpatient visits (mental health, substance use, medicine, primary care, emergency department, other, total). Each service utilization outcome was aggregated over the 1-year period following the date of Veterans' initial MISSION-Vet session.

## Data analysis

Our analytic strategy involved four components that align with the four study aims. Specifically, we examined: 1) pre-implementation organizational readiness; 2) IF process, including IF events; 3) MISSION-Vet implementation in the IU and IF time periods; and 4) association between MISSION-Vet and VA health services. Because the number of trained providers to deliver MISSION-Vet and the number of Veterans who received it at the seven homeless programs was too small for meaningful program comparison, our analysis focuses on a comparison between the two VAMCs (Sites A and B) rather than the seven individual programs when making comparisons for all measures of interest. In other words, practical considerations necessitated that we modify our intended analysis plan to make the site, rather than the program the cluster unit of interest.

First, we examined organizational readiness using descriptive statistics and conducted comparisons of organizational readiness between the Sites A and B and by staff type (case manager vs. peer specialist), staff age, staff sex, and duration of employment with the VA using nonparametric Wilcoxon and Kruskal-Wallis tests. Second, we used descriptive statistics to examine IF events, including number, duration, and type of IF activities. Third, and similarly, we

used descriptive statistics to examine implementation of MISSION-Vet. We summarize information about the number and type of MISSION-Vet sessions provided overall, at the Veteran-level, and by VAMC. We also examined provision of MISSION-Vet separately by staff type (i.e., whether a case manager or peer specialist). Additionally, to assess the potential impact of IF on MISSION-Vet, we examined how the overall provision of MISSION-Vet changed over time both before and after the start of IF using descriptive measures of the number of MISSION-Vet sessions provided at each site by month. Our intent was to estimate the intervention effect using a statistical model in line with established practices for stepped-wedge designs. However, this analysis plan was not feasible for two reasons. First, as noted above, the number of trained providers deliver MISSION-Vet and the number of Veterans receiving MISSION-Vet at each of the seven original program sites was small, which caused us to shift from using the program to the site as our cluster of interest in our analysis. Second, because neither of the two sites provided any MISSION-Vet services in the IU period, there was no variation in the outcome of interest during this time thus rendering it impractical to estimate such a model. We therefore rely solely on descriptive statistics to examine the impact of IF on the provision of MISSION-Vet services.

Fourth, as an exploratory analysis, we examined the relationship between receipt of MISSION-Vet and Veteran-level measures of engagement in clinical service (i.e., VA inpatient and outpatient services). To do so, we estimated a series of bivariate linear regression models in which our service utilization measures (i.e., number of outpatient and inpatient visits, by type, in the year after a Veteran's initial MISSION-Vet session) served as the outcomes of interest and the number of MISSION-Vet sessions in the year following a Veteran's initial MISSION-Vet session served as the predictor of interest in all models. As these models were purely exploratory, they did not adjust for any additional covariates.

## Results

### Staff characteristics

This study commenced in February 2016 and recruitment stopped in July 2019. 108 staff were trained in MISSION-Vet as part of Implementation as Usual across two VAMCs (93 at Site A across 11 trainings,15 at Site B across four trainings). Following training, 77 staff (69% Site A, 87% Site B) completed an organizational readiness and demographic survey. Most respondents were case managers (77%) or peer specialists (16%) and few were unknown (8%). There was an even mix of males (47%) and females (45%), and unknown (8%); the average age was 49 ± 12 (range 26–72), length of time in VA was 5 ± 5 years (range <1 year-25 years). It is noteworthy that while MISSION-Vet was intended to be implemented by a case manager-peer specialist dyad, peer specialists were less available at Site A, which is also why fewer peer specialists were trained throughout the project.

### Organizational readiness

As described in Table 1, staff reported moderate to high organizational culture and climate at both sites. Site B had higher scores than Site A on nearly all individual items, although only several differences were statistically significant, perhaps due to the small sample size. The most consistent differences between sites were on the ORCA context staff culture subscale, with 100% of Site B staff agreeing to cooperate to maintain and improve patient care effectiveness and being willing innovate to improve clinical procedures, compared to 85% ($p$ = .055) and 77% ($p$ = .03) at Site A, respectively. Sites also differed on implementation climate scores regarding support to use MISSION-Vet being higher at Site B than Site A (92% vs 76%, $p$ = .005), despite both sites reporting low recognition and appreciation for using MISSION-Vet.

Table 1. Organizational readiness scores by site, N = 77.

| | Site A (N = 64) | Site B (N = 13) | p |
|---|---|---|---|
| **Implementation Climate Score** | | | 0.181 |
| I am expected to use MISSION-Vet with a certain number of Veterans. | 65% | 75% | 0.313 |
| I am expected to help my organization meet its goals for implementing MISSION-Vet. | 79% | 77% | 0.984 |
| I will get the support I need to identify potentially eligible Veterans for MISSION-Vet. | 76% | 92% | 0.005 |
| I will get the support I need to use MISSION-Vet with Veterans. | 77% | 85% | 0.212 |
| I will receive recognition when I use MISSION-Vet with Veterans. | 52% | 58% | 0.845 |
| I will receive appreciation when I use MISSION-Vet with Veterans. | 49% | 58% | 0.638 |
| **ORCA/Context, Staff Culture, Staff** | | | 0.081 |
| Have a sense of personal responsibility for improving patient care and outcomes. | 82% | 100% | 0.099 |
| Cooperate to maintain and improve effectiveness of patient care. | 85% | 100% | 0.055 |
| Are willing to innovate and/or experiment to improve clinical procedures. | 77% | 100% | 0.028 |
| Are receptive to change in clinical processes. | 78% | 85% | 0.464 |
| **ORCA/Context, Leadership Culture** | | | 0.507 |
| Reward clinical innovation and creativity to improve patient care. | 70% | 91% | 0.075 |
| Solicit opinions of clinical staff regarding decisions about patient care. | 75% | 75% | 0.715 |
| Seek ways to improve patient education and increase patient participation in treatment. | 79% | 83% | 0.437 |
| **ORCA/Context, Leadership Behavior** | | | 0.418 |
| Provide effective management for continuous improvement of patient care. | 69% | 85% | 0.486 |
| Clearly define areas of responsibility and authority for clinical managers and staff. | 69% | 85% | 0.262 |
| Promote team building to solve clinical care problems. | 71% | 77% | 0.506 |
| Promote communication among clinical services and units. | 72% | 85% | 0.822 |
| **ORCA/Context, Measurement** | | | 0.304 |
| Provide staff with information on VA performance measures and guidelines. | 70% | 100% | 0.058 |
| Establish clear goals for patient care processes and outcomes. | 77% | 92% | 0.448 |
| Provide staff members with feedback/data on effects of clinical decisions. | 67% | 75% | 0.378 |
| Hold staff members accountable for achieving results. | 69% | 77% | 0.958 |

There were no statistically significant differences in organizational measures by role, sex, age, or site. There was, however, a positive significant relationship ($r = .32$, $p = .01$) between duration of VA employment and the ORCA measurement scale regarding goals, guidelines, feedback, and accountability.

## Implementation facilitation

Following Implementation as Usual, one external facilitator per site supported MISSION-Vet implementation. While we recognize that on the surface, it might suggest an imbalance in workload given that Site A was much larger than Site B, the facilitator in Site A had more available time to devote to this project. Facilitation activities included stakeholder engagement, site assessment, preparation/planning, ongoing process monitoring, education, program adaptation, marketing, and problem identification and problem solving. A total of 110 facilitation events averaging 27-minutes were conducted. At Site A, there was a total of 70 virtual (phone

or Skype) external facilitation events across the four participating programs at Site A that averaged 24 minutes over a 17-month period. There was a total of 40 facilitation events across the three participating programs at Site B averaging 34-minutes over an 11-month period, including 60% virtual and 40% in-person. Site B had a higher per person facilitation dose (i.e., how many staff participated in MISSION-Vet vs. how many facilitation events were provided) than Site A.

## Implementation as usual versus implementation facilitation

To explore the primary study objective, which is the potential impact of IF on provision of MISSION-Vet services, we examined how the overall provision of MISSION-Vet across both sites changed over time following the start date of IF. As noted above, although the calendar date on which IF began at each site varied, we standardized our analysis of MISSION-Vet over time to examine the provision of MISSION-Vet services over a standard 21-month period indexed to the start date of IF by site.

Fig 2 shows the results of this analysis and aggregates the total number of MISSION-Vet sessions provided across both Sites in the 6-months prior to and 12-months following the start of IF. There were no MISSION-Vet services provided at either site prior to the start of IF. However, MISSION-Vet services commenced immediately after the start of IF, suggesting that IF was effective in increasing the implementation of MISSION-Vet.

Table 2 summarizes results of our analysis of the provision of MISSION-Vet, including the between site comparisons of the provision of MISSION-Vet during the IF period. As the table shows, when examining the actual implementation of MISSION-Vet in the IF period, of the 108 staff trained, 53% at Site B tried MISSION-Vet, as evidenced by staff entering at least one MISSION-Vet note, as compared to 14% of staff at Site A completed a MISSION-Vet note. This difference was statistically significant ($p$ = .002). A total of 574 MISSION-Vet notes were entered during the IF period. This included 424 DRT notes (273 at Site A vs 151 at Site B) and 89 peer notes (32 at Site A vs 57 at Site B) by 14 case managers (9 at Site A vs 5 at Site B) and seven peer specialists (four at Site A vs three at Site B), with 93 (70 at Site A vs 23 at Site B) Veterans. Relatively fewer peer support sessions were conducted at Site A (12% of all sessions), as compared to Site B (38% of all sessions), as evidenced by the notes. Nearly all the community events were conducted at Site B (39 vs 2). No service referrals were conducted at Site A, compared to 114 at Site B.

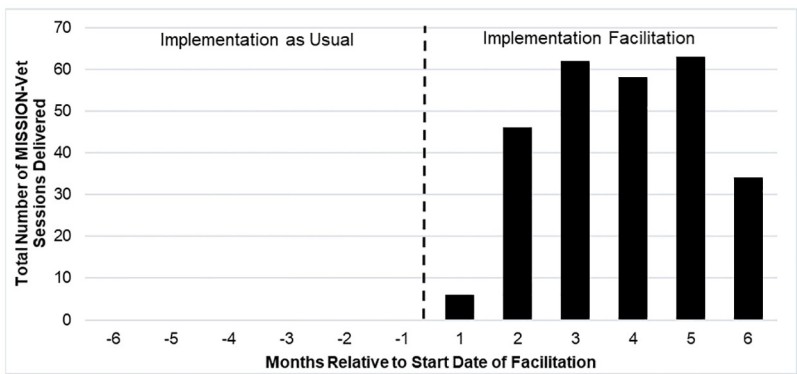

**Fig 2. Total number of MISSION-Vet sessions delivered across 2 VAMCs by month relative to the start of facilitation.**

**Table 2. MISSION-Vet implementation outcomes.**

| | Total | | Site A | | Site B | | *p* |
|---|---|---|---|---|---|---|---|
| | n | % | n | % | n | % | |
| **Staff Trained** | 108 | 100% | 93 | 100% | 15 | 100% | - - |
| Staff Implemented MISSION | 21 | 19% | 13 | 14% | 8 | 53% | 0.002 |
| Case Managers | 14 | 13% | 9 | 10% | 5 | 33% | - - |
| Peers | 7 | 6% | 4 | 4% | 3 | 20% | - - |
| **MISSION-Vet Implemented** | 734 | 100% | 348 | 100% | 386 | 100% | < .001 |
| DRT Sessions | 424 | 58% | 273 | 78% | 151 | 39% | - - |
| Peer Sessions | 89 | 12% | 32 | 9% | 57 | 15% | - - |
| Self-guided Exercises | 66 | 9% | 41 | 12% | 25 | 6% | - - |
| Community Events | 41 | 6% | 2 | 1% | 39 | 10% | - - |
| Service Referrals | 114 | 16% | 0 | 0% | 114 | 30% | - - |
| **Veterans served** | 93 | 100% | 70 | 100% | 23 | 100% | - - |
| DRT & Peer Session (Fidelity) | 17 | 18% | 3 | 4% | 14 | 61% | < .001 |
| DRT Session Only | 66 | 71% | 57 | 81% | 9 | 39% | - - |
| Peer Session Only | 2 | 2% | 2 | 3% | 0 | 0% | - - |
| Self-guided Exercises Only | 8 | 9% | 8 | 11% | 0 | 0% | - - |
| **Total Sessions per person (mean)** | 6.3 | - - | 4.9 | - - | 10.9 | - - | 0.005 |
| DRT Sessions/person (mean) | 5.4 | - - | 4.5 | - - | 7.9 | - - | 0.017 |
| Peer Sessions/person (mean) | 4.7 | - - | 6.4 | - - | 4.1 | - - | 0.426 |

With regards to the Veterans served, 93 received MISSION-Vet services by the trained staff during the IF period. Again, Site B provided more per-person DRT sessions, on average, (4.5 sessions, vs 7.9 sessions, *p* = .017) and similar per-person peer sessions, on average, (6.4 vs 4.1, *p* = .426) than Site A. No Veterans received the complete DRT dose or peer dose: 62 (67%) received only DRT sessions, two (2%) received only peer sessions, 17 (18%) received both DRT and peer sessions, and 12 (13%) received unstructured MISSION-Vet. MISSION-Vet was implemented with great overall fidelity with regards to both DRT and peer support at Site B as compared to Site A (61% vs 4%, *p* < .001). Furthermore, across Sites A and B, Veterans overall tended to remain engaged in MISSION-Vet, averaging 4.5-months of MISSION-Vet services of the anticipated 6-months of services.

## Health service utilization

The regression models we estimated to assess the relationship between the number of MISSION-Vet sessions and our health services utilization measures identified significant positive associations between the total number of MISSION-Vet sessions and the total number of outpatient medical-specialty visits, with each additional MISSION-Vet session associated with, on average, an additional 0.28 visits (B = .28, *t*(91) = 2.63, p = .001) in the one-year period after the start of MISSION-Vet services. Likewise, each additional MISSION-Vet session was associated with an increase of roughly 1 outpatient substance abuse treatment visit (B = .92, *t*(91) = 2.61, *p* = .01) and 2 total outpatient visits (B = 2.18, *t*(91) = 2.25, *p* = .03) in the one-year period following a Veteran's initial MISSION-Vet session. The number of MISSION-Vet sessions was not significantly associated with any of our inpatient service utilization measures, or the total number of inpatient hospitalizations.

## Discussion

In this hybrid III trial, we used a modified stepped-wedge design to compare IU versus IF with respect to the provision of MISSION-Vet. Our key findings were that neither site implemented MISSION-Vet with usual implementation and that IF significantly increased uptake of MISSION-Vet. Further, after IF, significantly more staff at Site B implemented MISSION-Vet compared to Site A. We also examined associations between pre-implementation organizational readiness and implementation. While we knew that Site B was a smaller and less complex site than Site A, we did not know the extent to which complexity and organizational readiness would drive outcomes. We found that the smaller site (Site B) had more flexibility to adopt MISSION-Vet than Site A. This suggests that it is not only the size of a site but also the organizational characteristics affected uptake. Higher MISSION-Vet uptake was associated with higher doses of IF overall, and this may be impacted by the in-person or virtual delivery. In exploratory analysis, this study also identified a positive relationship between the number of MISSION-Vet sessions and outpatient service utilization. These findings may suggest that MISSION-Vet assisted with outpatient treatment engagement, although some caution is warranted in making any firm conclusions about the extent to which this was the case, as this analysis did not adjust for any potential individual level or site-level factors that might confound the relationship between volume of receipt of MISSION-Vet and service utilization.

This project was done during a period of heightened attention to Veteran homelessness and following a commitment from the President of the United States and VA Secretary to end Veteran Homelessness [47]. It is not surprising that regarding organizational readiness, staff felt a personal responsibility for enhancing patient care and were interested in innovative practices to improve outcomes. However, these attributes alone were insufficient to stimulate uptake of MISSION-Vet. As mentioned above, several organizational characteristics were related to MISSION-Vet outcomes. We found that although Sites A and B had similar perceptions of leadership culture, staff culture, and implementation climate, Site B had a more supportive climate and staff were more willing to improve current practices. Powell et al. had similar findings that strategic (implementation climate) compared to general (organizational culture and climate, transformational leadership) organizational factors had more influence on knowledge and attitudes towards current mental health evidence-based practices [48]. Closer examination of how practitioners' perspectives of context and climate shift between pre-implementation (i.e., knowledge and attitudes) to implementation may offer important clues into the tailoring of facilitation and selection of other implementation strategies [40].

Consistent with other recent work on implementation of behavioral health care interventions, we determined IF could be an appropriate strategy to support staff behavior change and address baseline needs by sites, and specifically, interpersonal support tailored to local needs, the mechanism underlying IF, which may be of benefit [38, 39, 49]. As noted above, sites implemented MISSION-Vet only after IF was initiated as training alone did no stimulate MISSION-Vet use. On-site facilitation (Site B) appeared to be more effective in generating use compared to virtual support (Site A). Although Site B had fewer facilitation events, Site B's facilitation was delivered both in-person and virtually, and dose/personal attention was higher and more focused. These results also align with our prior MISSION-Vet study using GTO as the implementation strategy which also found that implementation support was needed to initiate MISSION-Vet use [18]. However, we do not attribute the differences only to virtual IF, as in reality, a constellation of contextual factors are responsible for the greater uptake and also include, medical center size and program scope and organizational readiness; these bear further study.

It is also interesting to note that no Veterans received a full dose of MISSION-Vet (24 sessions along with outreach and linkages as needed). Other studies of multicomponent behavioral health interventions have found implementation fidelity difficult to achieve compared to the implementation of singe component behavioral interventions and one provider [11, 12, 37, 50, 51]. Recent literature on facilitation indicates that it has a mixed legacy in implementation work; in some other studies facilitation is not as effective as the literature suggests in terms of format (active versus passive implementation strategies) and dose, however, for MISSION-Vet facilitation was a relatively productive approach [52, 53]. The contrast between Site A and B illustrates another aspect of this issue which is that larger medical centers may face additional barriers in implementing multimodal team-based behavioral interventions. However, among those Veterans offered MISSION-Vet, more MISSION-Vet services were associated with increased VA outpatient treatment engagement. Our prior studies have also found a relationship between MISSION-Vet, increased use of outpatient services and a reduction in homelessness [54]. Engagement in both MISSION-Vet and other outpatient treatments is critical for homeless Veterans with a co-occurring disorder as these Veterans often cycle in and out of healthcare services and are unstably engaged, which can result is housing instability and loss and exacerbation of medical problems [54].

Despite the promising findings, this study has several limitations. First, it was done in two VAMCs, of incomparable size, thus the data is not generalizable to the entire VA system. Importantly, practical considerations regarding the volume of MISSION-Vet provided across our original seven programs required that we adjust our clustering unit to the VAMC site. Second, given that no Veterans received a full dosed of MISSION-Vet, it was not possible to assess differences across the intended seven individual programs within VAMCs and thus we had to report comparison between the two VAMCs (Sites A and B). Third, this study used an IU comparison group and a slightly more active low intensity implementation intervention could have shown some effect. Nonetheless, this study did teach us that training alone (IU) was not effective in these VA settings and more proactive strategies (such as IF) may be needed to help guide future implementation. Fourth, IF was offered in person and virtually at Site B and virtually alone in site A, and this could have also been responsible for the site differences above and beyond the site size and organizational readiness differences, which were unable to be teased apart. Fifth, despite access to the VA medical record, no data was available regarding other clinical outcomes besides engagement as the VA data tracking client progress are not offered for every client at a standard time, thus making other clinical comparisons impossible. Sixth, it was not possible to control for secular changes over time between sites. Therefore, it is plausible that the increased national attention to the problem of homelessness during the implementation period or some other extraneous factor could account for the higher readiness and cooperation in Site B. However, while this is unlikely as no site did MISSION during IU, a qualitative component would have added additional nuance to the analyses.

## Conclusion

Despite these limitations, this study offers the field lessons on how to get a team-based multi-modal intervention like MISSION-Vet incorporated in both smaller and larger VA's and in programs serving a population of homeless Veterans with a co-occurring disorder. This study suggests that standard MISSION-Vet web training is insufficient, and a more active and ongoing implementation support is needed. It is also possible that some tailoring of MISSION-Vet within certain settings might be warranted in future implementation studies involving MISSION-Vet given the time and intensity needed for service delivery. Moreover, MISSION-Vet was specifically developed to offer a comprehensive service delivery experience rather than

Veterans receiving these services from separate programs and providers, the latter of which could fragment care. However, in the future, it might be feasible to cross train staff in programs already delivering MISSION-Vet type services such as assertive community outreach to deliver the psychoeducational components of MISSION-Vet or to train staff offering co-occurring disorders psychoeducational groups how to deliver community outreach and linkage support. Another important future direction might be to enhance the MISSION-Vet training, select a more intensive implementation strategy, as well as perhaps have a more robust external plus internal facilitation approach, the latter of which was previously found to be successful in implementing other complex clinical innovations in diverse healthcare settings [55]. Lastly, future studies might also examine virtual versus in-person facilitation in more rigorous ways.

## Acknowledgments

The views and opinions of authors expressed herein do not necessarily state or reflect those of the United States Government, and shall not be used for advertising or product endorsement purposes. We wish to thank the VA Central Office, Homeless Program Office for their support and guidance on this project and the seven homeless programs within the two VAMCs for the collaboration.

## Author Contributions

**Conceptualization:** David A. Smelson, Vera Yakovchenko, Thomas Byrne, Megan B. McCullough, Jeffrey L. Smith, Sonya Gabrielian.

**Data curation:** Thomas Byrne.

**Formal analysis:** Thomas Byrne.

**Funding acquisition:** David A. Smelson.

**Investigation:** David A. Smelson, Vera Yakovchenko, Megan B. McCullough, Jeffrey L. Smith, Kathryn E. Bruzios, Sonya Gabrielian.

**Methodology:** David A. Smelson, Vera Yakovchenko, Thomas Byrne, Megan B. McCullough, Jeffrey L. Smith.

**Project administration:** David A. Smelson, Megan B. McCullough, Kathryn E. Bruzios, Sonya Gabrielian.

**Resources:** David A. Smelson, Vera Yakovchenko, Jeffrey L. Smith, Kathryn E. Bruzios.

**Supervision:** David A. Smelson, Vera Yakovchenko, Jeffrey L. Smith, Sonya Gabrielian.

**Validation:** David A. Smelson, Vera Yakovchenko, Thomas Byrne.

**Visualization:** Vera Yakovchenko, Thomas Byrne.

**Writing – original draft:** David A. Smelson, Vera Yakovchenko, Thomas Byrne, Megan B. McCullough.

**Writing – review & editing:** David A. Smelson, Vera Yakovchenko, Thomas Byrne, Megan B. McCullough, Jeffrey L. Smith, Kathryn E. Bruzios, Sonya Gabrielian.

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
