## [Decision Letter · Decision Letter 0]

21 Oct 2021

PONE-D-21-13578Testing Implementation Facilitation for Uptake of an Evidence-Based Psychosocial intervention in VA Homeless Programs: A Hybrid Type III TrialPLOS ONE

Dear Dr. Bruzios,

Thank you for submitting your manuscript to PLOS ONE. After careful consideration, we feel that it has merit but does not fully meet PLOS ONE’s publication criteria as it currently stands. Therefore, we invite you to submit a revised version of the manuscript that addresses the points raised during the review process.

Please see detailed notes below based on 2 peer reviews.

We look forward to receiving your revised manuscript.

Kind regards,

Annika C. Sweetland, DrPH, MSW

Academic Editor

PLOS ONE

Journal Requirements:

2. Thank you for including your ethics statement: "The project was deemed quality improvement and received an exempt status by the Institutional Review Board at the Bedford, Massachusetts VAMC according to the VA Program Guide 1200.21 "

a) Please provide additional details regarding participant consent. In the ethics statement in the Methods and online submission information, please ensure that you have specified (1) whether consent was informed and (2) what type you obtained (for instance, written or verbal, and if verbal, how it was documented and witnessed). If your study included minors, state whether you obtained consent from parents or guardians. If the need for consent was waived by the ethics committee, please include this information.

3. You indicated that ethical approval was not necessary for your study. We understand that the framework for ethical oversight requirements for studies of this type may differ depending on the setting and we would appreciate some further clarification regarding your research. Could you please provide further details on why your study is exempt from the need for approval and confirmation from your institutional review board or research ethics committee (e.g., in the form of a letter or email correspondence) that ethics review was not necessary for this study? Please include a copy of the correspondence as an ""Other"" file.

Additional Editor Comments (if provided):

Our sincere apologies about the delayed decision on this manuscript. Due to difficulty finding an additional external reviewer, Academic Editor performed the secondary review.

Overall, I agree with Reviewer 1 that the article is interesting, well written, of value to the field and appropriate for publication in PLOS-One. However, some methodological issues raise questions about the interpretation of findings.

In addition to the concerns raised by Reviewer 1, I would add that a major concern is that the authors describe the study as having used a stepped-wedge design, but the analysis of findings does not match this. The authors describe pragmatic sequential implementation roll out occurring in 7 waves across two sites (site A followed by site B), wherein the IU in all sites (“control condition”) produced no change in any of the sites/waves. Since it was not possible to compare each wave to itself pre-implementation (intervention IF vs. control IU) as intended (suggested by the stepped-wedge design), instead the analysis shifts to a comparison of implementation outcomes between sites A and B, that have some significant qualitative differences (e.g. size, location, readiness) as well as implementation differences (e.g. hybrid in-person and virtual vs. virtual only). The findings are still valuable and interesting, but the analysis and conclusions need to match the methodological reality.

An additional limitation is that since the authors did not randomize the sequence of implementation across sites (all waves in site A followed by all waves in site B), it is not possible to control for secular changes over time between sites. It seems plausible that increased national attention to the problem of homelessness during the implementation period (lines 390-392) could be a factor that accounted for the higher readiness and cooperation in Site B.

Finally (minor) it may be worth highlighting that the current "training as usual" (IU) strategy at the VA of watching a self-instructional video was totally ineffectual should not be continued. A more proactive strategy (such as IF) could help guide more effective future implementation within the VA.

Reviewers' comments:

Reviewer's Responses to Questions

**Comments to the Author**

1. Is the manuscript technically sound, and do the data support the conclusions?

Reviewer #1: Partly

2. Has the statistical analysis been performed appropriately and rigorously? 

Reviewer #1: Yes

3. Have the authors made all data underlying the findings in their manuscript fully available?

Reviewer #1: Yes

4. Is the manuscript presented in an intelligible fashion and written in standard English?

Reviewer #1: Yes

5. Review Comments to the Author

Reviewer #1: This is an interesting paper that examines the implementation and uptake of the MISSION-Vet program for homeless veterans with co-occurring mental health and substance use disorders. The paper addresses knowledge gaps surrounding the optimal approaches to implementing multicomponent interventions like MISSION-Vet and the role of implementation facilitation in this process. The paper is well-written but I do have some concerns about some of the conclusions drawn by the authors based on their approach and findings.

Main comments:

- The authors have conducted a stepped wedge trial unlike other stepped wedge trials that I have seen. Normally with this design, all study sites are exposed to the control condition during the first time period and then with each passing time period a new site receives the intervention. Sites are randomized such that they will receive both control and intervention conditions but in different sequences. This is not what these authors have done, as each program received the same six months of implementation as usual followed by implementation facilitation for a six-month period. The intervention was rolled out in a step-wise fashion but the time in which sites were exposed to the control condition (“IU”) did not differ across programs or sites. This reality raises a number of questions that have implications for the conclusions that can be drawn about the influence of facilitation on implementation and other outcomes. My first question is what justified this atypical approach to data collection if the goal was to conduct a stepped wedge trial that captures secular trends in implementation?

- Given the atypical study design, it is hard to know whether the changes in number of MISSION-Vet services is truly due to the arrival of implementation facilitation, or could it be that a certain amount of time is necessary for sites to prepare for uptake of the intervention and that services would have been started to be delivered after 6 months regardless of the presence of facilitation. How do we know that the observed results are actually due to facilitation and not due to the normal time it takes to internally get organized and be ready to deliver new services? The inability to capture temporal trends in the study seems like a major limitation.

- In a similar vein, the authors argue that Site B may have certain characteristics (e.g. support for identifying veterans, willingness to innovate) that may have made it more receptive and ready for the MISSION-Vet program than in Site A. However, as per Figure 1, Site B also received its facilitation after all the programs in Site A had been exposed to facilitation. How do we know that the facilitation at Site B wasn’t significantly better given all the lessons learned through interactions with programs in Site A? Given also that the facilitation at site B included an in-person component, can we really draw firm conclusions about the role of program/site characteristics when there may have been more important differences at the level of the facilitation programs/sites received?

- One thing that was less clear to me was the number of facilitators involved in the project. The authors state on page 16 “one external facilitator per site supported MISSION-Vet implementation”. Does this mean that there were two facilitators in total, one for Site A and one for Site B? Or were there facilitators at each program site? If there were only two facilitators overall, this raises questions about how dedicated they were to each program. In Site A, the overlapping exposure to facilitation means that the facilitator would have had to provide supports to multiple programs at the same time. This does not appear to be the case for Site B, where the IF periods don’t overlap. Is it possible that the quality of facilitation differed because there were more competing demands on the facilitator providing supports in Site A?

- With respect to the linear regression analyses used in the study, the data are clearly in a hierarchical structure but I saw no attempts to determine whether multi-level analyses were feasible/appropriate or not. Also, it was not clear to me whether the regression analyses included any confounding variables, this should be made explicit. I would urge the authors to be cautious in their interpretation of results if no confounding variables were included in their models.

- As a reader, it remains unclear what explains the differences in intervention uptake across the different programs and sites. A qualitative component to this study would have been highly valuable but was not performed. This should probably be mentioned as a limitation because it is hard to draw conclusions based on the limited organizational readiness data, especially with the limited sample size at Site B.

Minor comments:

- On page 12, line 266, the authors state that neither of the two sites provided MISSION-Vet services during the “IF period” but it was during the “IU period” that no services were provided.

6. PLOS authors have the option to publish the peer review history of their article (what does this mean?). If published, this will include your full peer review and any attached files.

Reviewer #1: **Yes: **Matthew Menear

---

## [Author Response · Author response to Decision Letter 0]

29 Dec 2021

November 15, 2021

To Whom It May Concern,

We wish to thank the reviewers for their thoughtful feedback on our manuscript entitled “Testing implementation facilitation for uptake of an evidence-based psychosocial intervention in VA homeless programs: A hybrid type III trial and Manuscript # PONE-D-21-13578.” Below are the reviewer responses followed by ours in bold. They also include page numbers where they can be found in the manuscript.

Response to Reviewers

Journal Requirements:

This has been reviewed and addressed. 

2. Thank you for including your ethics statement: "The project was deemed quality improvement and received an exempt status by the Institutional Review Board at the Bedford, Massachusetts VAMC according to the VA Program Guide 1200.21"

a. Please provide additional details regarding participant consent. In the ethics statement in the Methods and online submission information, please ensure that you have specified (1) whether consent was informed and (2) what type you obtained (for instance, written or verbal, and if verbal, how it was documented and witnessed). If your study included minors, state whether you obtained consent from parents or guardians. If the need for consent was waived by the ethics committee, please include this information.

We have now provided additional details regarding participant consent in the Methods section. On page 7, lines 155-160 we explain “…thus waiving the need for written or verbal informed consent. The staff being trained were the study participants and they were informed about the project’s designation as Quality Improvement. Data from the medical record were extracted for the clients being served by the staff members offering MISSION-Vet. Staff were informed of this data extraction and records were not anonymized as it was necessary to identify the clients being served by the staff offering MISSION-Vet.

As noted above, the IRB deemed this Quality Improvement and thus waived the requirement for informed consent. It is also noteworthy that we did not include anonymized records as we needed to identify the clients being served by the staff offering MISSION-Vet. See page 7, lines 155-160.

As noted above, this statement was added to the Methods section on page 7 lines 155-160. It reads “…thus waiving the need for written or verbal informed consent. The staff being trained were the study participants and they were informed about the project’s designation as Quality Improvement. Data from the medical record were extracted for the clients being served by the staff members offering MISSION-Vet. Staff were informed of this data extraction and records were not anonymized as it was necessary to identify the clients being served by the staff offering MISSION-Vet.” This has also been added to the ethics statement field in the submission form.

3. You indicated that ethical approval was not necessary for your study. We understand that the framework for ethical oversight requirements for studies of this type may differ depending on the setting and we would appreciate some further clarification regarding your research. Could you please provide further details on why your study is exempt from the need for approval and confirmation from your institutional review board or research ethics committee (e.g., in the form of a letter or email correspondence) that ethics review was not necessary for this study? Please include a copy of the correspondence as an ""Other"" file.

This has been addressed in comments 2a and 2b and within the manuscript (see page 7 lines 155-160). A copy of the memo determining that the IRB called this non-research has been uploaded as an “Other” file. 

Thank you for bringing this to our attention. The phrase has been removed from the manuscript.

Additional Editor Comments:

5. Overall, I agree with Reviewer 1 that the article is interesting, well written, of value to the field and appropriate for publication in PLOS-One. However, some methodological issues raise questions about the interpretation of findings.

In addition to the concerns raised by Reviewer 1, I would add that a major concern is that the authors describe the study as having used a stepped-wedge design, but the analysis of findings does not match this. The authors describe pragmatic sequential implementation roll out occurring in 7 waves across two sites (site A followed by site B), wherein the IU in all sites (“control condition”) produced no change in any of the sites/waves. Since it was not possible to compare each wave to itself pre-implementation (intervention IF vs. control IU) as intended (suggested by the stepped-wedge design), instead the analysis shifts to a comparison of implementation outcomes between sites A and B, that have some significant qualitative differences (e.g. size, location, readiness) as well as implementation differences (e.g. hybrid in-person and virtual vs. virtual only). The findings are still valuable and interesting, but the analysis and conclusions need to match the methodological reality.

Thank you for this feedback. Because stepped-wedge design (comment #8), intervention groups (comment #13), and analysis (comment #12) come up separately in reviewer 1’s feedback, we address these below for ease of review and to reduce duplication. In addition, we address contextual and facilitation differences below (comment #9, 10, 11).

6. An additional limitation is that since the authors did not randomize the sequence of implementation across sites (all waves in site A followed by all waves in site B), it is not possible to control for secular changes over time between sites. It seems plausible that increased national attention to the problem of homelessness during the implementation period (lines 390-392) could be a factor that accounted for the higher readiness and cooperation in Site B.

We appreciate this feedback and have added the following statement in the limitations described in the discussion on page 25, lines 495-500 about secular trends. We note “Sixth, it was not possible to control for secular changes over time between sites. Therefore, it is plausible that the increased national attention to the problem of homelessness during the implementation period or some other extraneous factor could account for the higher readiness and cooperation in Site B. However, while this is unlikely as no site did MISSION during IU, a qualitative component would have added additional nuance to the analyses.” 

7. Finally (minor) it may be worth highlighting that the current "training as usual" (IU) strategy at the VA of watching a self-instructional video was totally ineffectual and should not be continued. A more proactive strategy (such as IF) could help guide more effective future implementation within the VA. 

We appreciate this feedback and have now added this more boldly in the discussion on page 23, line 449. We note “As noted above, sites implemented MISSION-Vet only after IF was initiated as training alone did no stimulate MISSION-Vet use.” We also added this as a limitation on page 25, lines 487-489, “Nonetheless, this study did teach us that training alone (IU) was not effective in these VA settings and more proactive strategies (such as IF) may be needed to help guide future implementation.”

Reviewers' comments:

Reviewer #1: 

Main comments:

8. The authors have conducted a stepped wedge trial unlike other stepped wedge trials that I have seen. Normally with this design, all study sites are exposed to the control condition during the first time period and then with each passing time period a new site receives the intervention. Sites are randomized such that they will receive both control and intervention conditions but in different sequences. This is not what these authors have done, as each program received the same six months of implementation as usual followed by implementation facilitation for a six-month period. The intervention was rolled out in a step-wise fashion but the time in which sites were exposed to the control condition (“IU”) did not differ across programs or sites. This reality raises a number of questions that have implications for the conclusions that can be drawn about the influence of facilitation on implementation and other outcomes. My first question is what justified this atypical approach to data collection if the goal was to conduct a stepped wedge trial that captures secular trends in implementation?

Thank you for the thoughtful feedback on the design. We agree that the description of our stepped-wedge trial lacks clarity. We have changed the manuscript in a number of places to more clearly describe the design of our study. First, we now call it a “modified stepped-wedge” design in the Abstract (line 35), pages 6-7, and 21. We have also included citations for these modified stepped wedge design (reference #s 23, 24). Second, besides characterizing it as a modified step wedge design, on 6-7, lines 132-148 we further explain it. We write “This is a modified stepped-wedge design in that, in contrast to a traditional stepped-wedge wherein all sites initially start in the control condition, with programs switching over to the intervention condition at fixed intervals or steps and then remaining in the intervention condition for the duration of the study, in our study, it was not feasible to implement IF at all programs simultaneously. As a result, in the present study, each program received six months of IU and crossed over to IF for another six months, with data collection continuing for an additional six months after the end of IF. Moreover, in terms of calendar time, the rollout of the control and intervention conditions in our study occurred in a sequential manner across programs, with each program being in the intervention and control conditions for a fixed period of time, and not contributing to the analysis in periods in which they are neither in the intervention or control condition (Fig 1). This type of design is akin to what the stepped-wedge design literature refers to as an incomplete stepped-wedge design with limited measurements prior to and after crossover [23, 24] and is appropriate to use when it is not feasible to implement an intervention or collect data in all clusters simultaneously, either due to resource constraints or for other reasons. In this case, we did not have the resources to do IF in every location at the same time.” In this case, we did not have the resources to do IF in every location at the same time. Third, we now have more clearly depicted this in Figure 1 on page 7, line 52.

9. Given the atypical study design, it is hard to know whether the changes in number of MISSION-Vet services is truly due to the arrival of implementation facilitation, or could it be that a certain amount of time is necessary for sites to prepare for uptake of the intervention and that services would have been started to be delivered after 6 months regardless of the presence of facilitation. How do we know that the observed results are actually due to facilitation and not due to the normal time it takes to internally get organized and be ready to deliver new services? The inability to capture temporal trends in the study seems like a major limitation.

Thank you for bringing up this important point about differences in facilitation and potential organizational differences related to MISSION-Vet uptake. We have added two statements to address this in the discussion section. First, on page 22, line 418-420, “Higher MISSION-Vet uptake was associated with higher doses of IF overall, and this may be impacted by the in-person or virtual delivery.” Then we added the following limitation on page 25, lines 495-500, “Sixth, it was not possible to control for secular changes over time between sites. Therefore, it is plausible that the increased national attention to the problem of homelessness during the implementation period or some other extraneous factor could account for the higher readiness and cooperation in Site B. However, while this is unlikely as no site did MISSION during IU, a qualitative component would have added additional nuance to the analyses.” 

10. In a similar vein, the authors argue that Site B may have certain characteristics (e.g. support for identifying veterans, willingness to innovate) that may have made it more receptive and ready for the MISSION-Vet program than in Site A. However, as per Figure 1, Site B also received its facilitation after all the programs in Site A had been exposed to facilitation. How do we know that the facilitation at Site B wasn’t significantly better given all the lessons learned through interactions with programs in Site A? Given also that the facilitation at site B included an in-person component, can we really draw firm conclusions about the role of program/site characteristics when there may have been more important differences at the level of the facilitation programs/sites received?

We appreciate this point about differences in facilitation delivery. We now mention the in-person vs. virtual facilitation in the discussion section on page 22, lines 418-420, “Higher MISSION-Vet uptake was associated with higher doses of IF overall, and this may be impacted by the in-person or virtual delivery.”

11. One thing that was less clear to me was the number of facilitators involved in the project. The authors state on page 16 “one external facilitator per site supported MISSION-Vet implementation”. Does this mean that there were two facilitators in total, one for Site A and one for Site B? Or were there facilitators at each program site? If there were only two facilitators overall, this raises questions about how dedicated they were to each program. In Site A, the overlapping exposure to facilitation means that the facilitator would have had to provide supports to multiple programs at the same time. This does not appear to be the case for Site B, where the IF periods don’t overlap. Is it possible that the quality of facilitation differed because there were more competing demands on the facilitator providing supports in Site A?

Yes, this study included one facilitator per site. Thank you for also noting that there may have been differences in facilitator quality between the two sites and their respective facilitators. We have clarified this on page 17, lines 337-339, “While we recognize that on the surface, it might suggest an imbalance in workload given that Site A was much larger than Site B, the facilitator in Site A had more available time to devote to this project.”

12. With respect to the linear regression analyses used in the study, the data are clearly in a hierarchical structure but I saw no attempts to determine whether multi-level analyses were feasible/appropriate or not. Also, it was not clear to me whether the regression analyses included any confounding variables, this should be made explicit. I would urge the authors to be cautious in their interpretation of results if no confounding variables were included in their models.

We thank the reviewer for this comment about the analysis. We consider these analyses to be exploratory in nature and have clarified this in a number of places in the manuscript. In the data analysis plan, we added on page 12, lines 271-273, “In other words, practical considerations necessitated that we modify our intended analysis plan to make the site, rather than the program the cluster unit of interest.” We also added on page 13, lines 288-292, “However, this analysis plan was not feasible for two reasons. First, as noted above, the number of trained providers deliver MISSION-Vet and the number of Veterans receiving MISSION-Vet at each of the seven original program sites was small, which caused us to shift from using the program to the site as our cluster of interest in our analysis.” We also added clarification to this statement on page 13, line 294-296, “We therefore rely solely on descriptive statistics to examine the impact of IF on the provision of MISSION-Vet services.” On page 13, line 297, we explain “as an exploratory analysis,” and thus we did not attempt to account for clustering by site and do not include confounders, which we also added on page 14, lines 304-305; “As these models were purely exploratory, they did not adjust for any additional covariates.” This is also clarified on page 22, line 420-421, “In exploratory analysis, this study also identified a positive relationship between the number of MISSION-Vet sessions and outpatient service utilization.” We also agree that, given the limitations of this analysis, it is important to be cautious in interpreting the results of these models. Our interpretations are now more clearly articulated on page 22, lines 421-426, “These findings may suggest that MISSION-Vet assisted with outpatient treatment engagement, although some caution is warranted in making any firm conclusions about the extent to which this was the case, as this analysis did not adjust for any potential individual level or site-level factors that might confound the relationship between volume of receipt of MISSION-Vet and service utilization.” In the discussion section we now more clearly articulate the limitations of this analysis where we added the following statement on page 25, lines 480-482, “Importantly, practical considerations regarding the volume of MISSION-Vet provided across our original seven programs required that we adjust our clustering unit to the VAMC site.”

We also note “exploratory analysis” on page 22. 

13. As a reader, it remains unclear what explains the differences in intervention uptake across the different programs and sites. A qualitative component to this study would have been highly valuable but was not performed. This should probably be mentioned as a limitation because it is hard to draw conclusions based on the limited organizational readiness data, especially with the limited sample size at Site B.

Thank you, we agree that a qualitative component would add nuance to the analyses. We address this limitation on page 25, lines 495-500, “Sixth, it was not possible to control for secular changes over time between sites. Therefore, it is plausible that the increased national attention to the problem of homelessness during the implementation period or some other extraneous factor could account for the higher readiness and cooperation in Site B. However, while this is unlikely as no site did MISSION during IU, a qualitative component would have added additional nuance to the analyses.” However, while this is unlikely as no site did MISSION during IU, a qualitative component would have added additional nuance to the analyses. 

Minor comments:

14. On page 12, line 266, the authors state that neither of the two sites provided MISSION-Vet services during the “IF period” but it was during the “IU period” that no services were provided.

Sorry for this confusion. This is now changed to “IU” on page 13, line 293.

Thank you again for your review and for your consideration of this manuscript. Please address all correspondence concerning this manuscript to me at David.Smelson@va.gov.

Dr. David A. Smelson, PsyD

Professor 

Director of the University of Massachusetts Center of Excellence in Addictions

Department of Psychiatry, University of Massachusetts Medical School

---

## [Decision Letter · Decision Letter 1]

2 Mar 2022

Testing Implementation Facilitation for Uptake of an Evidence-Based Psychosocial Intervention in VA Homeless Programs: A Hybrid Type III Trial

PONE-D-21-13578R1

Dear Dr. Bruzios,

We’re pleased to inform you that your manuscript has been judged scientifically suitable for publication and will be formally accepted for publication once it meets all outstanding technical requirements.

Kind regards,

Annika C. Sweetland, DrPH, MSW

Academic Editor

PLOS ONE

Additional Editor Comments (optional):

Reviewers' comments:

Reviewer's Responses to Questions

**Comments to the Author**

1. If the authors have adequately addressed your comments raised in a previous round of review and you feel that this manuscript is now acceptable for publication, you may indicate that here to bypass the “Comments to the Author” section, enter your conflict of interest statement in the “Confidential to Editor” section, and submit your "Accept" recommendation.

Reviewer #1: All comments have been addressed

2. Is the manuscript technically sound, and do the data support the conclusions?

Reviewer #1: Yes

3. Has the statistical analysis been performed appropriately and rigorously? 

Reviewer #1: Yes

4. Have the authors made all data underlying the findings in their manuscript fully available?

Reviewer #1: Yes

5. Is the manuscript presented in an intelligible fashion and written in standard English?

Reviewer #1: Yes

6. Review Comments to the Author

Reviewer #1: The authors have addressed all of my comments and I am satisfied with their responses, I have no further comments.

7. PLOS authors have the option to publish the peer review history of their article (what does this mean?). If published, this will include your full peer review and any attached files.

Reviewer #1: **Yes: **Matthew Menear

---

## [Editor Report · Acceptance letter]

7 Mar 2022

PONE-D-21-13578R1 

Testing Implementation Facilitation for Uptake of an Evidence-Based Psychosocial Intervention in VA Homeless Programs: A Hybrid Type III Trial 

Dear Dr. Bruzios:

I'm pleased to inform you that your manuscript has been deemed suitable for publication in PLOS ONE. Congratulations! Your manuscript is now with our production department. 

Kind regards, 

on behalf of

Dr. Annika C. Sweetland 

Academic Editor

PLOS ONE